# OpenReview forum: "ContextCite: Attributing Model Generation to Context"
_NeurIPS.cc/2024/Conference — NeurIPS 2024 poster_

### Official Review · Reviewer_4MHt · 2024-06-14

**Soundness:** 3
**Presentation:** 3
**Contribution:** 3
**Rating:** 6
**Confidence:** 4

**Summary:**

This paper proposes a simple and effective approach for attributing LLM generation to the sources in the context, where a set of random ablations and their corresponding effects in the model's probability of generating the response are modeled with a (sparse) linear relationship. The method archives impressive performance and also allows applications such as verifying model generations or denoising the sources in the context.

**Strengths:**

- The problem of attribution is very important, and the paper focuses on an understudied aspect of how different sources *cause* the model to generate what it generates.
- The proposed method is simple and intuitive. Comparison with the baselines is thorough and clearly demonstrates its superior effectiveness.
- The paper is generally well-written and easy to follow.

**Weaknesses:**

- The method may be a bit costly to run, given that the linear relationship needs to be re-learned for every (context, generation) instance, and the number of sample ablations also grows when the context scales.
- If I'm not mistaken, the proposed linear relationship implicitly assumes that the sources in the context contribute in a rather independent manner to the model's generation. This may not work well when the generation involves complex interactions and reasoning over a large number of sources in the context (beyond just a few sentences as in, e.g., HotpotQA). For example, if the task involves some kind of logical deduction, missing a certain premise in the context could cause a snow-balling effect on how the model processes the remaining sources and scores its generation, which is very hard to model with a linear relation.

**Questions:**

For extracting query-relevant information from the context (Section 5.2), if some important sources are "lost in the middle" and overlooked by the model, then the proposed approach would give a very low score for them as they don't contribute to model generation. Then they also won't get selected during the extraction phase. How would this improve QA performance then?

**Limitations:**

Yes.

---

> ### Author Rebuttal · Authors · 2024-08-06
>
> We thank the reviewer for their feedback. We address each question individually below.
>
> **Effect of number of context sources (and sparsity) on sample complexity.**
>
> We agree with the reviewer that computational cost is a significant limitation of ContextCite. However, we would like to point out that the number of sample ablations needed often does not actually grow quickly as the context scales. As we discuss in Appendix C.1.1, the number of ablations required to learn an accurate surrogate model only grows linearly in the number of *relevant* sources (rather than growing linearly in the total number of sources). In Figure 3(a), we observe empirically that the number of relevant sentences often remains small ($<10$) even when the context consists of hundreds of sources. This suggests that a small number of context ablations may suffice even for very long contexts (in practice, we find that using just $32$ context ablations often works well).
>
> We would also like to point out that while ContextCite is expensive to run, it can be applied in a post-hoc manner. A user can interact with an LLM as usual and can only choose to apply ContextCite and incur this cost in cases when they are suspicious of a model’s generated response and would like to see attributions.
>
> **Effect of linearity in context attribution.**
>
> We agree with the reviewer that a linear model may not handle complex dependencies between sources, such as those that might arise when reasoning over many sources. We do find it promising that a linear model works well for Hotpot QA, in which every question explicitly requires combining information from two or more documents. That said, proper attribution for more sophisticated reasoning may require more complex surrogate models (or interaction terms for the linear model), which is an interesting direction for future work.
>
> **Why does selecting only relevant sources help?**
>
> The reviewer makes a good point that if the model uses the wrong sources to generate a response, then selecting just these sources is unlikely to help. To understand why relevant sources can actually help, we consider two failure modes associated with long contexts:
> 1. The model identifies the wrong sources. Selecting these sources will *not* help.
> 1. The model identifies the correct sources, but *misinterprets* information because it is distracted by other irrelevant information. Selecting these sources *can* help because a shorter context including these sources could help.
>
> We intend to include this discussion of why selecting relevant sources helps in the next revision.

---

> > ### Comment · Reviewer_4MHt · 2024-08-10
> >
> > Thank you for the response, which partially addresses some of my concerns. I'll keep my earlier evaluations.

---

### Official Review · Reviewer_A2Ps · 2024-07-14

**Soundness:** 2
**Presentation:** 3
**Contribution:** 3
**Rating:** 7
**Confidence:** 3

**Summary:**

The current paper introduces the task of context attribution, which aims at attributing a generated response of an auto-regressive LM back to the sentences in the input contexts. Target at this task, ContextCite, was proposed to predict which piece of context changes the probability of the generated response most. Together with ContextCite, this paper also designed two evaluation metrics based on the probabilities of the generated responses. At length, the experiments on two small LLMs show the effectiveness of ContextCite. Additionally, this paper provides several examples of applications of ContextCite.

**Strengths:**

- This paper introduces the task of context attribution, which, AFAIK, is the first time this important and interesting task has been introduced. The task is well-motivated and well-defined. One could expect that this paper could initiate a new line of work on interpretability.
- Along with the proposed task, this paper proposed a solution, namely ContextCite, building on a simple but effective idea. The subsequent evaluation suggested its promising performance.
- One thing that I like the most about this work is that in addition to the task and the model, this paper also introduces two well-designed evaluation metrics as well as a list of potential applications of ContextCite.

**Weaknesses:**

Though I thought this paper was already of good quality, I still have three major concerns regarding the task, the model and the evaluation.

First and foremost, in my opinion, the current definition of the task is still highly limited and can be largely extended. For example, the current task only focuses on the attributions with responses to the whole generated responses, but for tasks like summarisation (which was also examined in this work), it is also important to attribute each token in the response to, for instance, route the hallucination. I am personally very OK if the solution of such an enhanced task is not provided, but I am very inclined to see that the related discussions about the possible extensions of the task can be included.

Second, regarding ContextCite, though the idea of making use of the probability change looks promising, after due consideration, I have a strong feeling that its relation with which parts in the input contribute more to the response is not fully deterministic. In my opinion, the probability of a response could be reduced more if, as pointed out in the paper, the omitted piece contributes more to the response or only if the resulting context after removing the sentence is way more incoherent. One example of such a possibility is given in Appendix C2, but the discussion seems not to be sufficient. This also made me think that the model only suits tasks whose output relies on the very long input text (e.g., summarisation and document-based QA, as omitting one sentence would not highly influence the coherency).

Finally, the evaluation was done on three very related tasks and two tiny language models. This makes me somewhat question whether the solution is generalizable to other NLP tasks (that are open-ended, for example), other larger models and other prompt designs (e.g., in-context learning and chain-of-thought).

**Questions:**

Can you provide some examples of the selected sentences and hit on the distributions and characteristics of the selected sentences?

**Limitations:**

See my points in the weakness section.

---

> ### Author Rebuttal · Authors · 2024-08-06
>
> We thank the reviewer for their feedback. We address each question individually below.
>
> **Token-level context attribution.**
>
> Our task actually does consider attributing arbitrary selections from the response (including individual tokens). We discuss this in Section 2.3. When we evaluate ContextCite in Section 4, we evaluate the attributions of individual sentences in the response: “given an example, we (1) split the response into sentences using an off-the-shelf tokenizer and (2) compute attribution scores for each sentence”; see lines 202-203. We have also included a figure in the global response illustrating an example of a ContextCite attribution on Llama-3-70B for a particular generated statement.
>
> **Effect of ablations on context coherence and model output.**
>
> The reviewer makes a good point that a context source might not directly contribute to a response, but ablating it might influence the response indirectly by making the context incoherent. For example, consider the sentences: “John lives in Boston. Charlie lives in New York. He sometimes visits San Francisco.” In this case, “He” refers to Charlie. However, if we ablate just the sentence about Charlie, “He” will now refer to “John.” So, if we attribute the generated statement “Charlie sometimes visits San Francisco,” then we’ll get two sources: (1) “Charlie lives in New York” and (2) “He sometimes visits San Francisco.” While, by some definition, (2) is the actual source, having both (1) and (2) as sources is also reasonable behavior in our view. So, we do not view this issue as a significant limitation. Extending ContextCite to account for such dependencies (e.g., via a non-linear surrogate model) is an interesting avenue for future work.
>
> **Experiments on larger models and other NLP tasks.**
>
> - **Larger models:** Our experiments in the paper evaluate ContextCite on models ranging from 4B to 8B parameters. To showcase the scalability of our method, we have included an additional figure in the global response, which illustrates a ContextCite attribution for Llama-3-70B. Furthermore, in Appendix B.3, we show that our experiment from Section 5.2 (selecting query-relevant context sources) improves results on LLama-3-70B as well.
> - **Other NLP tasks:** The tasks we use to evaluate ContextCite are reasonably diverse, including summarization (CNN DailyMail), knowledge extraction (TyDi QA), and reasoning using information from multiple documents (Hotpot QA). Still, we agree with the reviewer that certain tasks/prompt designs may be less suitable for ContextCite. As we note in Appendix C.2, if the model outputs a chain-of-thought reasoning chain, then it is quite possible that only the initial facts would yield proper attributions. For example, for the output “I have 3 eggs. You have 4 eggs. Together, we have 3 + 4 = 7 eggs.”, ContextCite would be able to meaningfully attribution “I have 3 eggs” and “You have 4 eggs”, but not “Together, we have 3 + 4 = 7 eggs” because this statement follows from the previous part of the response and not directly from the context. Extending the core task of context attribution to chain-of-thought style reasoning chains is an interesting avenue for future work.
>
> **Examples of attributions.**
>
> We’ve provided an example attribution in the global response, and will include additional examples in the next revision.

---

> > ### Comment · Reviewer_A2Ps · 2024-08-09
> > **Thanks for your response.**
> >
> > Thanks for your response. Though I still think what I wrote in my review are still limitations of this work, I agree they are minor points. I'll keep my recommendation of an acceptance.

---

### Official Review · Reviewer_Thhj · 2024-07-21

**Soundness:** 3
**Presentation:** 4
**Contribution:** 2
**Rating:** 5
**Confidence:** 4

**Summary:**

In an attempt to understand how language models leverage context information in its generation, this work studies contributive context attribution. Following the recent trend in attribution research, authors propose to use (sparse) linear models to learn the importance of each unit/sentence in the context, and demonstrate its effectiveness using top-k log-probability drop and linear datamodeling score (LDS) tests. Lastly, authors explore two potential applications of contributive context attribution, namely generated statement verification and query-relevant information selection.

**Strengths:**

The paper is very clearly written, well-structured, and easy to follow:

- The problem of contributive context attribution is relatively new. However, the authors provided all the necessary contexts for the problem, and thus I was able to understand the problem setup easily.
- ContextCite seems to achieve promising attribution accuracy on both top-k log-probability drop and LDS tests, especially in comparison to baseline methods.
- Their context selection experiment (section 5.2) results also look promising. If we make the analogy between data attribution and context attribution, this probably mirrors a data selection experiment. It's interesting to see that context selection generally leads to the performance improvement.

**Weaknesses:**

- While (contributive) context attribution looks to be new on the surface, their problem setup, method (i.e. ContextCite), and evaluation are mostly borrowed from existing data attribution literature. While authors provided proper references to those literature, I still believe the technical contribution largely lacks novelty. I wasn't able to grasp whether authors introduced any major modifications to adapt existing data attribution methods (i.e. datamodel) and evaluations (i.e. brittleness and LDS) to context attribution. If I missed something, please correct me.
- If my understanding is correct, they prompted GPT-4 to generate the (proxy) gold label for their generated statement verification experiments (section 5.1). This was confusing because the authors noted in line 84-85 that prompting the model for citations is generally considered as corroborative attribution. To be precise, the authors asked for the correctness of each sentence instead of for citations here, but it was a bit odd for me to use the prompting technique to evaluate contributive context attribution.

**Questions:**

- How does the context length affect the number of required forward passes with randomly sampled context information? Naively thinking, if you have n parameters in your linear model, you may need n samples to properly fit the linear model (without a sparsity assumption).

**Limitations:**

The authors adequately addressed the limitations in Appendix C.

---

> ### Author Rebuttal · Authors · 2024-08-06
>
> We thank the reviewer for their feedback. We address each question individually below.
>
> **On technical novelty and connections to data attribution.**
>
> We agree with the reviewer that our work leverages the data attribution literature (e.g., datamodels, LDS); we acknowledge and discuss these connections in our paper. That said, understanding how a model uses information presented in its context is different (both conceptually and mechanically) from understanding how a model uses its training examples. Furthermore, some of the applications enabled by context attribution (e.g., helping verify correctness, see Section 5.1) don’t have clear data attribution analogues. Hence, despite the connections, we believe that it is valuable to study context attribution as its own task.
>
> We also believe that it is a valuable empirical finding that a linear surrogate faithfully models how a language model uses its context (especially given the dependencies between sentences). A priori it is not clear that this design choice from the data attribution literature would work well for context attribution too.
>
> **Confusion over corroborative attribution and usage of GPT-4.**
>
> We would like to briefly clarify: the goal in Section 5.1 is to use the contributive attributions estimated via ContextCite to help verify the correctness of a generated statement. Our intuition is that if the contributive sources (i.e., sources which *cause* a model to generate a statement) do not also *entail* the generated statement, then the generated statement might be incorrect (because the model may have misinterpreted the sources or hallucinated). We use GPT-4 only to assess whether each generated statement is actually correct and not for any sort of attribution.
>
> The reviewer is right that verifying correctness is closely related to corroborative attribution. In Section 5.1 though, we only focus on exploring whether *contributive* attribution methods (like ContextCite) can help in verifying the correctness of generated statements.
>
> **Effect of number of context sources on sample complexity.**
>
> The reviewer makes a good point that in general, if we have $n$ sources, we require $O(n)$ forward passes to learn an accurate linear surrogate model. As we discuss in Appendix C.1.1, in the case where the ground-truth model is sparse, however, we actually only require $O(k\log(n))$ forward passes where $k$ is the number of non-zero entries in the ground-truth model. In Figure 3(a), we illustrate empirically that even when the context is very long (hundreds of sources), the number of *relevant* sources (i.e., sources whose removal causes a probability drop beyond some threshold) is generally small ($<10$). As a result, in practice, just $32$ forward passes are enough to learn an accurate surrogate model even when there are hundreds of sources.

---

### Official Review · Reviewer_zvan · 2024-07-26

**Soundness:** 3
**Presentation:** 4
**Contribution:** 3
**Rating:** 8
**Confidence:** 3

**Summary:**

The authors of 17808 formalizes the problem of context attribution for language models. That is, identifying which parts of the input context caused a model to generate a particular output. The authors propose ContextCite, a method that learns a sparse linear surrogate model to approximate how ablating different parts of the context can impact the model's output probability. Authors claim several contributions, including 1) Formalizing the official task of context attribution and evaluation metrics, 2) Developing ContextCite, a simple and scalable attribution method, 3) Demonstrating ContextCite outperforms baselines on attribution quality, and 4) Showing applications in verifying generated statements and improving response quality.

**Strengths:**

- Important problem formulation: Context attribution addresses a key challenge in understanding and improving llm behavior / use cases esp. faithfulness and interpretability.
- Comprehensive experiments: Right metric, datasets, and models. Comprehensive experiments are provided.
- Case study -- Two valuable downstream applications are addressed - verifying generated statements (5.1) and improving response quality (5.2).
- Clear writing and presentation -- Overall I find the presentation of the paper is well-structured and easy-to-follow.

**Weaknesses:**

- As noted in Appendix C.2, the current ablation approach of removing sentences can break dependencies between sents. This limitation could be explored a bit further in the paper, but it's understandable if there's time constraint.

- A little more theoretical justification for why the linear surrogate model works well would be better.

**Questions:**

- While the efficiency and simplicity of linear surrogate models are acknowledged, did you try more sophisticated surrogate models beyond linear regression in your early exps? If so, how did they look like?

**Limitations:**

The authors provide a good discussion of limitations in Appendix C.2.

---

> ### Author Rebuttal · Authors · 2024-08-06
>
> We thank the reviewer for their feedback. We address each question individually below.
>
> **Justification for linear surrogate modeling.**
>
> Our justification for using linear surrogate models for context attribution is purely empirical. A priori, we do not see any theoretical reason why linearity is the “right” design choice for context attribution. That said, across NLP tasks (summarization, knowledge extraction, reasoning over multiple documents), we empirically observe that a linear surrogate faithfully models how language models use their context (i.e., the surrogate model’s predictions match the actual log-probability drops, see Appendix B.1). Furthermore, by inducing sparsity, we are able to learn accurate linear surrogate models efficiently (with just $32$ context ablations, even when the context consists of hundreds of sources).
>
> **On non-linear surrogate modeling and sentence-level dependencies.**
>
> We agree that extending ContextCite to work with non-linear surrogate models is a promising avenue for future work. As the reviewer points out, surrogate model classes that factor in interactions among context sources (e.g., decision trees, linear models with interaction terms) could naturally account for the dependencies between context sources. Moreover, the expressivity of non-linear context attribution could help in capturing the effect of context ablations on model generations more accurately. In this work, however, we did not look into nonlinear variants of ContextCite for two reasons:
> First, as per our evaluation in Section 4, sparse linear surrogate models already result in accurate and sample-efficient context attributions. More sophisticated surrogate models may require many more context ablations to learn accurately.
> Second, linear models directly yield an attribution score for each source, making them interpretable. It would be less clear how to assign an attribution score to each source if the surrogate model were, say, a decision tree.

---

### Author Rebuttal · Authors · 2024-08-06

We thank all reviewers for their helpful feedback, which we think have highlighted areas where we could have been clearer. We have responded to each reviewer individually in detail, and we use this comment to highlight the additional experiments we have conducted in our appendices as well as in response to reviewer concerns.

**Paper contributions.**

We would like to reiterate the three main contributions of this work:
1. Introduce the task of context attribution, i.e., understanding how a language model uses information provided in its context when responding to a query.
1. Present ContextCite, a simple and scalable method for context attribution.
1. Explore two applications of context attribution: (i) help verifying factuality and (ii) selecting query-relevant information to improve response quality.

**Additional experiments in response to reviewers’ questions.**

In response to the reviewers’ feedback, we perform three new experiments, as outlined below. We will add these experiments to the next revision.
- Scaling ContextCite to large-scale language models (Llama-3-70B)
- Applying ContextCite with words as sources (instead of sentences) on a reasoning benchmark (DROP)
- A new application: detecting adversarial poisons in (backdoored) long contexts

**Attached PDF.**

We’ve also attached an example of a ContextCite attribution of Llama-3-70B for a randomly sampled example from the CNN DailyMail news summarization task. We include this example to illustrate that ContextCite scales to larger models and can be used to attribute arbitrary selections from the generated response (not just the whole response).

---

### Public Comment · ~Pietro_Lesci1 · 2025-05-13
**Missing bibliography**

Hi there,

While reading this paper, I noticed that the bibliography section is missing. I found the same issue on the version hosted on the NeurIPS website. I hope this helps!

Best,
Pietro

---

### Decision · Program_Chairs · 2024-09-25

**Decision:**

Accept (poster)

**Comment:**

This paper investigates attribution in generation. The main contributions of this paper include (1) initially introducing the task of context attribution; (2) proposing the simple and scalable method for context attribution, ContextCite; (3) showcasing two applications of context attribution, respectively verifying factuality and selecting query-relevant information to improve response quality.

This paper has several strengths. First, as all reviewers have agreed, this paper is well-structured and easy to follow. Second, this paper formalizes context attribution for language models, i.e., identifying which parts of the input resulted in generation of a particular output. In addition, reviewers (Thhj, A2Ps, and 4MHt) agreed that the proposed ContextCite is simple and intuitive, with superior performance over baselines. The experiments designed for method verification are comprehensive and convincible, as reviewers Zvan and A2Ps mentioned.

Several weaknesses were identified by reviewers. (1) There are some concerns regarding experiments. For example, reviewer A2Ps mentioned exploration of only limited number of tasks, LLMs, and prompts which needs to be extended; reviewer Zvan mentioned concerns regarding ablations. (2) The method seems indeed costly to run. (3) As reviewers pointed out, several in-depth aspects remain to be further analyzed, including adding theoretical justification, and discussion between context and data attribution.

During the rebuttal period, most of the major concerns seem to be addressed, including adding new experiment results in response to reviewers’ questions and discussing each argument in depth.